# Learning anomalies from graph: predicting compute node failures on HPC clusters

Jože M. Rožanec[*1], Roy Krumpak[1], Martin Molan[2], and Andrea Bartolini[2]

[1]Jožef Stefan Institute
[2]University of Bologna
 joze.rozanec@ijs.si, krumpak.roy@gmail.com, {martin.molan2, a.bartolini}@unibo.it

## Abstract

Today, high-performance computing (HPC) systems play a crucial role in advancing artificial intelligence. Nevertheless, the estimated global data center electricity consumption in 2022 was around 1% of the final global electricity demand. Therefore, as HPC systems advance towards Exascale computing, research is required to ensure their growth is sustainable and environmentally friendly. Data from infrastructure monitoring can be leveraged to predict downtimes, ensure these are treated in time, and increase the overall system's utilization. In this paper, we compare four machine-learning approaches, three of them based on graph embeddings, to predict compute node downtimes. The experiments were performed with data from Marconi 100, a tier-0 production supercomputer at CINECA in Bologna, Italy. Our results show that the machine learning models can accurately predict downtime, matching current state-of-the-art models.

## 1 Introduction

The increasing research and deployment of artificial intelligence require massive hardware, which has strong implications for HPC and data center energy sustainability and makes efficient utilization of the resources even more critical [1]. Maintaining consistent availability of HPC resources is crucial to avoid negative impact on research and minimize the carbon footprint as well [2]. Correctly forecasting potential compute node downtimes is critical to this end. Predictive maintenance has been shown to enable the efficient planning of maintenance tasks to minimize operation downtimes and preserve the health of the entire system [3].

Machine learning has shown great promise in providing accurate downtime predictions in HPC and data center environments to realize predictive maintenance. Among the works describing this approach, we find Pelaez et al. [4], who described how clustering was applied to perform online failure prediction. Klinkenberg et al. [5] followed a different approach, leveraging a supervised machine learning model trained on monitoring data to predict lock events. More recently, Borghesi et al. [6] developed deep learning models to predict compute node downtimes in HPC systems.

The fact that network architectures are governed by the same organizing principles regardless of the science domain and can provide a unified representation of heterogeneous data is a compelling reason driving research at the intersection of network science and machine learning [7].

The approaches described above leverage sensor readings or logs to predict compute node downtimes. Nevertheless, such approaches miss much contextual information by not being able to include additional points of view, such as the kind of information being monitored or the sensor placement. Such information can be included through a graph representation. E.g., Molan et al. [8, 9] joined sensor readings for each point in time considering the sensors' topological location within a particular rack. Nevertheless, such a representation provides no information on the types of sensors being used. Furthermore, the graph representations provide no context regarding past sensor readings - something that could be relevant to understanding whether we are heading to a particular state. To mitigate these issues, we propose a different graph representation with nodes representing specific sensor types and associated sensors and subgraphs resulting from translating time series into networks following specific heuristics. Such graphs are then encoded into embeddings and used for downstream compute node downtime prediction.

This work follows up research from Krumpak, Rožanec et al.[10] and focuses on understanding the information captured by Graph2Vec embeddings when considering time series data represented as Natural Visibility Graphs or enriched with contextual information regarding the sensor data (sensor ID and sensor type) and how these affect the predictive outcomes. We develop local and global machine-learning models that leverage graph embeddings summarizing domain knowledge regarding sensor types and readings to minimize system unavailability. We trained and evaluated the machine learning models on a subset of the publicly available data from the Marconi 100 supercomputer

---

*Corresponding author: Jože M. Rožanec: joze.rozanec@ijs.si

Proceedings of the 6th Northern Lights Deep Learning Conference (NLDL), PMLR 265, 2025.

compiled by Borghesi et al. [11]. The models' performance was evaluated considering the AUC ROC scores. Our results show that the models trained with graph embeddings and sensor reading information performed best. Furthermore, global models exhibited a stronger performance than local ones. While sensor reading data and graph embeddings resulted in a stronger mean performance than using sensor reading data only, the difference remained consistently small across the forecasting horizons.

The rest of this paper is structured as follows: Section 2 presents related work, Section 3 introduces the dataset we worked on. Section 4 describes our methodology and Section 5 the experiments performed. Finally, Section 6 presents and discusses the results obtained, and Section 7 provides our conclusions.

## 2 Related work

The size, complexity, and heterogeneous architecture of contemporary HPC systems require introducing machine learning methodologies that support the work of the system administrators [12]. As characterized by Netti et al. [12], anomaly detection and prediction are among the applications with the most direct impact on the overall availability, and by extension, sustainability [13], of the HPC system. Consequently, much effort has been invested in building data-driven models for anomaly detection and, later, anomaly prediction and anticipation. Various lines of research have been proposed. Considering the kind of data used to predict compute node failures, we could distinguish approaches that focus on log data and approaches that focus on sensor data. Among the ones focused on log data, Li et al. [14] explored how alerts containing a timestamp, a verbosity level, and a textual message describing the error could be leveraged to predict node failures. The authors achieved production-ready results, and their system has been deployed at Alibaba, reaching an AUC ROC score between 0.91 and 0.92 and lead times of 48 hours. In a similar line of research, Alharthi et al. [15] explored how sentiment analysis on log messages could be used to predict compute node issues. Furthermore, [16] explored a self-supervised approach to predict forthcoming log events, their location, and the expected lead time to failure.

Among the approaches that aim to predict node failures from telemetry data, we find RUAD [17] or PROCTOR [18] that leverage self-supervised learning and are only capable of recognizing the anomalies and failures that are taking place. They cannot provide failure predictions, allowing the system administrators to manage the system proactively, including intelligent scheduling considering anticipated hardware failures and preventative maintenance. While there has been an effort to create an anomaly antici-

pation system for HPC compute node failures, such as the one proposed by Borghesi et al. [19], such systems can anticipate the failure but provide no estimation about the time frame for it. Failure prediction approaches exist for components within the compute node, such as the work of Devesh Tiwari et al. [20] or Yu Liu et al. [21] focus on component failure prediction (disk failure specifically). Besides not being holistic and only covering a part of potential compute node failures, these approaches also have limited prediction windows [22].

The prediction model must include additional information beyond the telemetry data to go beyond component failure prediction or node-level failure detection. One such approach is GRAAFE [13], which is based on the observation that the additional information about the physical layout of the compute nodes within a computing room aids in the ability to train the compute node anomaly prediction model. This information is encoded as a graph: each compute node is represented as a vertex connected to its nearest neighbors, and node telemetry data is represented for each compute node as a vertex attribute. While different graph topologies have been tested, the optimal graph topology uncovered is a line graph representing a single compute rack in a compute room. Different approaches to anomaly detection with graphs have been applied to other domains and could be considered in HPC environments. Among them, we find anomaly detection with graph approaches that aim to alleviate structural distribution shifts and novel techniques for posing the problem as a temporal graph clustering problem [23, 24].

## 3 Dataset

The source of our data is a collection of the Marconi 100 supercomputer sensory data, which was gathered by Borghesi et al. and was made publicly accessible at https://zenodo.org/records/7541722 [11]. More specifically, we focused on a subset of the original data stored in the 1.tar distribution file where information about sixteen computing nodes in one rack of the system are available. The data is a two-dimensional table, where rows represent timestamps taken 15 minutes apart, ranging between March 9$^{th}$ 2020 and September 28$^{th}$ 2022. This means roughly 86 thousand rows for each compute node data. Approximately 9 thousand rows have missing values. Columns, on the other hand, contain different sensor measurements of the system aggregated over the 15 minute time interval, which include the power consumption of the fans and CPUs, the temperature of the GPUs cores and memory, the voltage of the power supply and many more. To be more precise, each sensor measurement is given in 4 columns, which store the minimum and maximum values, the

average values, and the standard deviation of the measurement. Two additional columns are provided, one that stores the time information and the other one that annotates anomalies in the data. The last column has integer values, where zero means there was no anomaly, and a value greater than zero signals there was an anomaly, which means the system was unavailable then. The timestamp column has date and time as values, and the remaining sensor columns have numerical values.

Accounting for all the sensors, there are 354 columns, though we chose only a small subset of the sensors to focus on. Our point of interest is the columns with the average values of the measurements. Furthermore, we selected some columns from which to get our data. These columns include ambient_avg, dimm0_temp_avg, fan_disk_power_avg, gpu0_core_temp_avg, gpu0_mem_temp_avg, p0_io_power_avg, p0_mem_power_avg, p0_power_avg, ps0_input_power_avg, ps0_output_curre_avg, ps0_output_volta_avg, fan0_0_avg and p0_vdd_temp_avg.

## 4  Methodology

**Data preprocessing** To develop the features needed for training our model, we followed a series of steps to preprocess the data and generate new representations of the information.

First, we addressed any missing values in the dataset by applying a forward-fill strategy, where the last observed value was carried forward to replace any missing entries within each column. The forward-fill strategy was chosen under the assumption that if a sensor did not emit a value or if a reading was lost, the most likely value to exist if conditions did not change in between, it would be the latest sensor reading.

Next, we utilized a change detection algorithm based on a relative moving average and a predefined threshold to identify significant changes in the data for each column. Once the indices indicating changes were identified, we replaced the values between two consecutive change points with the mean of that data segment. This process produced a simplified dataset with distinct segments of different values for each column. We exemplify such signal processing in Fig. 1.

To handle segments where values varied by a non-significant amount, we divided the range of values in each column into five quantiles and then mapped the values according to their respective quantiles. This quantization made the data discrete, resulting in each column containing only five possible values. The choice to perform discretization into five possible values was empirical weighting the number of states resulting from a particular number of selected bins. Consequently, there is a finite number

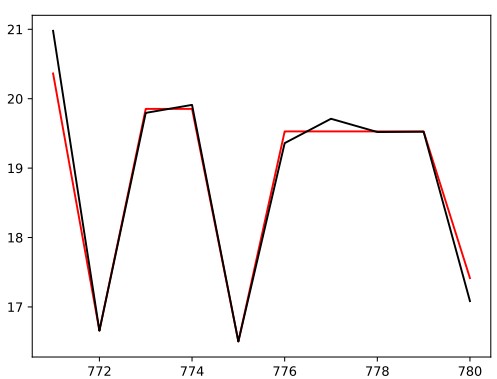

**Figure 1.** The Figure presents two signals: (i) an example of sensory data (black), and (ii) the post-processed signal once change detection and value replacement were performed (red).

of unique combinations of column values, which we interpreted as distinct states in the system.

**Feature engineering** Four different kind of features were created. First, given the compute node states identified above, we performed a one-hot encoding representation of this data. Nevertheless, given a one-hot encoded representation provides only insight into that specific state but no context on the preceding ones, we considered a more sophisticated approach. Given a time series of length n, natural visibility heuristics can be applied to create a graph $G = (V, E)$, where nodes $V$ represent time series data points and edges $E$ represent visibility relationships between those nodes. Then a deep learning model can be used to transform the graph $G$ into a vector representation $\mathbf{H} = f_{\text{embed}}(G)$. Choosing k=5 would encode information from each sensor with the same number of columns as the one-hot encoded approach while eventually conveying more information. To this end, we constructed natural visibility graphs for each sensor in a given state, considering the values of up to ten preceding states. In Fig. 2 we show how a sample drawing on how sensor signal was reduced into a sequence of unique values and later to a visibility graph. The Graph2Vec approach was used to transform the graphs into vector embedding representations.

Finally, we also created a slightly different graph representation encoding information about the sensors and joined it to the abovementioned visibility graphs to the nodes representing the sensors from which the sensor readings originate (see Fig. 2). This way, the random walks would convey information about current and past states and how these relate to particular sensors and sensed aspects (e.g., temperature). These graphs were then turned into an embedding using the Graph2Vec approach.

To create natural visibility graph embeddings, we

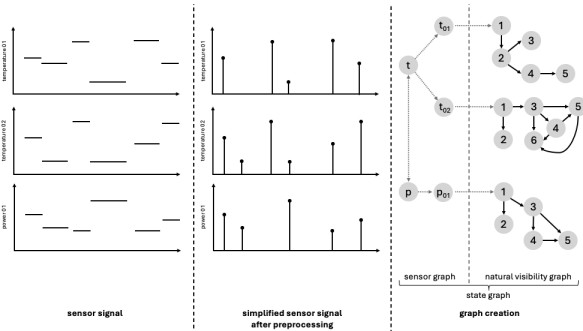

**Figure 2.** The Figure presents three signals and processing stages. The sensor signal is preprocessed considering change level detection and clustering into bins. The resulting values are then mapped into natural visibility graphs. The natural visibility graphs can be further enriched with information regarding the sensor they belong to and its type.

employed a Graph2Vec model trained over all of the visibility graphs, regardless of the sensor. The embedding creation process began with generating tagged random walks over a graph, which served as documents for training the Graph2Vec model. Once the model was trained, new random walks over the graphs were conducted, and these were used to perform inference and obtain a vector from the previously trained model. The final embedding for each graph was obtained by averaging these vectors, providing a robust representation of the graph's structure. Since one state produced several natural visibility graphs (for each sensor, there was one natural visibility graph), the resulting embeddings were concatenated into one vector embedding for that state, following the same positional convention we used for the one-hot encoding schema. In addition, we also created the state graph embeddings following the same procedure as with the natural visibility graph embeddings, with the difference being that instead of using the natural visibility graphs, we used the state graphs. Because the state graphs were larger we decided to perform a random walk of length 20 per node.

**Model training** We trained two kinds of CatBoost classifiers: (i) models trained on data from each compute node separately (local models) and (ii) models trained on data from all of the nodes (global model). In the case of local models, there were, on average, 8700 available unique state instances, which we used to train the model, considering a 75/5/20 split for train, validation, and testing. The splits were made such that the information about the time of each instance was preserved and that the training data was recorded before the validation data, and the validation data was recorded before the testing data. The global model's process was similar to that of the local models. First, we took the 75% training data from the local models and joined them into

one training data for the global model. The same was done with the validation set. When we tested the global model, we did so for each compute node separately to compare the results between the local models and the global model node by node.

The CatBoost classifier was trained running for 250 iterations with a cross-entropy loss, a learning rate of 0.1 and an L2 regularization factor of 0.3. The local and global models were trained for three forecast horizons determined by state changes. Considering the states changed every 165 minutes on average, the compute node downtimes could be predicted to up to 495 minutes ahead.

**Model evaluation** We measured the classifiers' discriminative performance with the AUC ROC metric. The score was computed at a compute node level and then summarized to report the mean, minimum (min), and maximum (max) values obtained across nodes for each experiment.

## 5 Experiments

We performed four experiments to understand how different features and graph representations affected the outcomes when predicting compute node downtimes. Our focus on graph representations is rooted in the fact that while much sensor data could be encoded by providing features that capture $t_{n-1}$ values, graph representation allows the conveying of information regarding the sensor time series and associating it to specific domain knowledge, such as sensor ID and sensor types. Three forecasting horizons were considered, predicting one to three states ahead. We detail them below:

**Experiment A** The purpose of the first experiment was to establish a baseline, considering the one-hot encoded representation of the compute node states. The feature vector size was 65.

**Experiment B** The purpose of the second experiment was to determine whether a natural visibility graph representation of past sensor values provides additional information that could enhance the models performance. The natural visibility graph representation was used to encode sensor data and a Graph2Vec representation to turn those graphs into embeddings. A single Graph2Vec model was used to learn visibility graphs from all the sensors. The decision was made under the assumption that the embedding model would learn best from a higher number and variety of samples than would have learned if exposed only to visibility graphs of a single sensor. To make the results comparable against *Experiment A*, we created embeddings for each sensor considering a vector size 5. Therefore, the final feature vector size matched the original one with a size of 65.

**Experiment C** The purpose of the third experiment was to determine whether the information encoded in experiments A and B was complementary and, therefore, joining it would lead to better results. The intuition behind the experiment was that (i) the one-hot encoded representation only had information about the current state but missed the sensor values observed in the near past, and (ii) the visibility graphs only encode the topology of a time series but do not encode information regarding the actual time series values. The feature vector was created by concatenating the hot encoded features with the natural visibility graph embeddings. The feature vectors had a total length of 120 values.

**Experiment D** The purpose of the fourth experiment was to determine whether the representation from Experiment C could be enriched with an additional embedding representation that not only considered the visibility graphs but also encode information about the aspects that they measured and the particular sensor that provided the sensor values. To that end, we created an additional graph representing the sensors, their sensing domain, and the associated natural visibility graphs. This experiment used the one-hot encoded data, the natural visibility graph embeddings, and the state graph embeddings. The state graph embeddings were generated with the Graph2Vec model and given a size of 15, resulting in an overall feature vector size of 135.

## 6 Results and discussion

Table 1 shows the results obtained across the compute nodes and breaks them down by experiment and whether a global or local model was trained. The results show that local models leveraging the one-hot encoded state features obtained the best mean results in most cases. We found these results deferred from the ones obtained at [10], achieving slightly worse results, and we could not pinpoint the exact cause of such a difference between executions. Using natural visibility graph embeddings alone resulted in poor performance. The addition of the state graph resulted in a marginal improvement in some cases. Nevertheless, further experimentation is required to draw conclusions in which cases do such embeddings help the model learn better and when they negatively affect the overall performance.

We have no explanation to why the graph embeddings alone displayed poor performance and further research is required to reach a better understanding. Among possible reasons we consider the fact that the natural visibility graphs capture only the topological aspects of the time series and any information related to their particular values is lost. This could be amended by adding complementary

**Table 1.** Results for the models in different experiments. The 'time' column indicates how many steps ahead from the current state the model tries to predict: $n+1$ means the model tries to predict the target variable of the next state, $n+2$ means the model tries to predict the target variable of two states ahead. The 'mean', 'min', and 'max' columns provide the aggregated results computed over the results obtained from individual computational nodes. The results are divided into 'local' and 'global' according to the data used to train the model.

| Exp. | Time | Local | | | Global | | |
|---|---|---|---|---|---|---|---|
| | | Mean | Min | Max | Mean | Min | Max |
| **A** | $n+1$ | 0.8085 | 0.7375 | 0.8831 | 0.7997 | 0.7270 | 0.8918 |
| | $n+2$ | 0.7844 | 0.7321 | 0.9193 | 0.7872 | 0.7056 | 0.9112 |
| | $n+3$ | 0.7533 | 0.6780 | 0.8531 | 0.7555 | 0.6937 | 0.8672 |
| **B** | $n+1$ | 0.5414 | 0.5029 | 0.5908 | 0.5538 | 0.5081 | 0.6041 |
| | $n+2$ | 0.5499 | 0.5119 | 0.5989 | 0.5726 | 0.5424 | 0.6183 |
| | $n+3$ | 0.5599 | 0.5034 | 0.6138 | 0.5751 | 0.5283 | 0.6166 |
| **C** | $n+1$ | 0.8070 | 0.7088 | 0.8857 | 0.7874 | 0.7037 | 0.8809 |
| | $n+2$ | 0.7707 | 0.6246 | 0.9144 | 0.7661 | 0.6953 | 0.9013 |
| | $n+3$ | 0.7281 | 0.6141 | 0.8211 | 0.7466 | 0.6757 | 0.8480 |
| **D** | $n+1$ | 0.7766 | 0.6284 | 0.8565 | 0.8086 | 0.7234 | 0.8866 |
| | $n+2$ | 0.7484 | 0.6008 | 0.9097 | 0.7915 | 0.7163 | 0.9024 |
| | $n+3$ | 0.7174 | 0.6608 | 0.8390 | 0.7566 | 0.6758 | 0.8674 |

graph representations such as quantile graphs or ordinal partition graphs. Another reason could be the quality of the Graph2Vec embeddings, which may depend on how the graph was sampled. A more thorough evaluation of hyperparameters is required to understand how the number of paths and length affect the outcomes.

## 7 Conclusion

Enabling predictive maintenance at HPC and datacenters is critical to minimize downtimes and maximize their utilization. Machine learning models have shown great promise accurately forecasting such events ahead of time. In this research we propose a graph-based approach to predicting compute nodes downtime and evaluate it on public data from Marconi 100 - a tier-0 production supercomputer from CINECA. To that end, we first process the sensor reading data by treating missing values and discretizing them. We used natural visibility graphs to represent time series and include them into graph representations describing compute nodes and their sensors. Such graphs were converted into embeddings through Graph2Vec models and used downstream to train a classifier and predict compute node downtimes. The results suggest that using graph embeddings could enhance the classifier performance in some cases. Nevertheless, further research is required to understand when such representations are beneficial or detrimental to the overall models' performance.

## Acknowledgments

This research was developed as part of the Graph-Massivizer project funded under the Horizon Europe research and innovation program of the European Union under grant agreement 101093202 https://graph-massivizer.eu/ and supported by the Slovenian Research Agency.

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
