# OpenReview forum: "Learning anomalies from graph: predicting compute node failures on HPC clusters"
_NLDL.org/2025/Conference — NLDL 2025 Oral_

### Official Review · Reviewer_AqLd · 2024-09-24
**Comment**

**Confidence:** 4

**Summary:**

The paper is about graph anomaly detection, which is an interesting topic. In this paper, authors aim to predict compute node failures on HPC clusters. The paper is well written and well organized. However, there are several concerns in the current version of the paper that addressing them will increase the quality of this paper.

**Strengths:**

1 Reasonable writing logic.

2 Novel ideas.

**Weaknesses:**

1 The authors should provide more background in the abstract to facilitate better understanding by readers outside the field.

2 The authors can consider showing more model structure figures and experimental result figures instead of expanding a simple figure.

3 In the methodology section, have you considered using mathematical formulas to outline and clearly express the model design?

4 Some related works can be considered.
[1] Deep Temporal Graph Clustering. ICLR 2024.
[2] Alleviating structural distribution shift in graph anomaly detection. WSDM 2023.

**Justification:**

The author considered the problem of graph anomaly detection in industrial scenarios, which is worthy of recognition, but there are still some concepts and details that are not clearly introduced.

---

> ### Author Rebuttal · Authors · 2024-10-25
>
> We thank the reviewer for the comments. Below, we provide a point-by-point answer to the questions they raised:
>
> (1) The authors should provide more background in the abstract to facilitate better understanding by readers outside the field.
>
>  We thank you for this observation. We have reworked the abstract with this observation in mind.
>
> (2) The authors can consider showing more model structure figures and experimental result figures instead of expanding a simple figure.
>
> We thank you for this observation. We have removed two Figures and included a new one providing better insights on how sensor data how is preprocessed and graphs created.
>
> (3) In the methodology section, have you considered using mathematical formulas to outline and clearly express the model design?
>
> We thank you for this observation. We have included some mathematical formulas to outline and express the model design more clearly.
>
> (4) Some related works can be considered. [1] Deep Temporal Graph Clustering. ICLR 2024. [2] Alleviating structural distribution shift in graph anomaly detection. WSDM 2023.
>
> We thank you for pointing us to these works. We have included them in the manuscript and will consider the approaches described in the abovementioned papers in future work as part of our research.

---

### Official Review · Reviewer_PR3L · 2024-09-27
**Review of the "Learning anomalies from graph" paper**

**Confidence:** 3

**Summary:**

This paper addresses the pressing challenge of predicting compute node downtimes in HPC systems, highlighting the importance of predictive maintenance for enhancing system sustainability and efficiency. The authors employ a data-driven approach using publicly available data from the Marconi 100 supercomputer, leveraging advanced ML techniques, particularly focusing on graph embeddings.

One of the significant strengths of the proposed methodology is the innovative representation of computing node data as natural visibility graphs and state graphs, which are subsequently transformed into embeddings. This approach is well-grounded in existing literature, demonstrating its relevance and potential for successful application in related fields, as evidenced by works such as "Understanding Graph Embedding Methods and Their Applications" by Xu (2021) and "Visibility Graph-Based Wireless Anomaly Detection for Digital Twin Edge Networks" by Bertalanic et al. (2024).

Regarding correctness, the preprocessing steps implemented by the authors appear appropriate and methodical. The forward-fill strategy for handling missing values, the application of a change detection algorithm, and the quantization of the data into discrete segments effectively prepare the dataset for subsequent modeling. Additionally, the shift from one-hot encoding to graph-based embeddings allows for the incorporation of historical context, enhancing the depth of information conveyed.

**Strengths:**

One of the strengths of this approach is the diversity of setups employed throughout the experimentation. By exploring various data preprocessing techniques (one-hot encoding, natural visibility graph to vector embeddings, the combination of the both of them and this combination plus state graph embedding) the study effectively addresses the complexity of anomaly detection in HPC environments. This variety not only enhances the robustness of the findings but also allows for a comprehensive evaluation of different methodologies, thereby providing valuable insights into the strengths and weaknesses of each setup. The willingness to experiment with multiple data transformations and modeling techniques demonstrates a thorough and thoughtful approach to tackling the challenges associated with anomaly detection, ultimately contributing to a deeper understanding of the problem space.

**Weaknesses:**

The proposed methodology is very interesting, but discussions about the experimental results are poor.

The discussion in sections 5 and 6 seems to suggest that the representation as vector embedding needs to be improved, without considering that perhaps the classifier was unable to take advantage of this representation. The poorer performance of CatBoost with natural visibility graph embeddings compared to simple one-hot encoded data may stem from several factors: (i) the transformation into natural visibility graphs and embeddings might have lost critical temporal or contextual information inherent in the original data. This could dilute the model's ability to recognize patterns associated with anomalies.

The embedding process may not have effectively captured the relationships or nuances present in the original dataset, leading to suboptimal representations; (ii) CatBoost is optimized for categorical features and may not perform as well with embeddings that require deeper contextual understanding. If the embeddings do not align well with the model's strengths, performance could suffer.

Alternative models, such as deep learning architectures (e.g., LSTM or CNNs), might have leveraged the embeddings more effectively due to their ability to capture complex patterns and relationships in high-dimensional data.

Table 2 should present in bold the best result for each metric in each predictive horizon.

Typo: sometimes it is written one-hot encoded, other times one hot encoded. It must standardize. On line 316 it just says "hot encoded"

**Justification:**

I assigned a score of 4 out of 5 for the acceptance of this paper due to its strong methodological approach and innovative use of graph-based embeddings. The proposed representation of compute node data as natural visibility graphs and state graphs offers significant advantages, as it enables the capture of complex relationships and temporal dependencies inherent in the data. This method enhances the richness of the features utilized for predicting compute node downtimes, contributing to a more nuanced understanding of the system's behavior.

However, the discussion of the experimental results is somewhat superficial. The authors could strengthen their analysis by providing a more in-depth comparison of the preprocessing setups and correlating the data representations with the topology of the classifiers used. Furthermore, the paper would benefit from a more explicit positioning within the existing literature on high-performance computing, particularly in relation to forecasting and anomaly detection methodologies. For example, insights from the validation techniques discussed in "Data-driven Deep-learning Forecasting for Oil Production and Pressure" by Werneck et al. (2022) could enhance the rigor of their validation approach, especially regarding multi-step-ahead predictions.

While the paper presents a noteworthy methodological innovation and the experiments conducted are intriguing, a deeper critical reflection on the results and a more robust connection to the state of the art would significantly improve its impact. Nonetheless, I support the approval of this paper due to its contributions to the field and its potential for further exploration and development.

---

> ### Author Rebuttal · Authors · 2024-10-25
>
> We thank the reviewer for the comments. Below we provide a point-by-pint answer to the questions they raised:
>
> (1) "However, the discussion of the experimental results is somewhat superficial. The authors could strengthen their analysis by providing a more in-depth comparison of the preprocessing setups and correlating the data representations with the topology of the classifiers used. Furthermore, the paper would benefit from a more explicit positioning within the existing literature on high-performance computing, particularly in relation to forecasting and anomaly detection methodologies. For example, insights from the validation techniques discussed in "Data-driven Deep-learning Forecasting for Oil Production and Pressure" by Werneck et al. (2022) could enhance the rigor of their validation approach, especially regarding multi-step-ahead predictions."
>
> We thank you for the observations above. We have reworked the related work section. We could not access the abovementioned paper but we will consider it in the future.
>
> (2) Alternative models, such as deep learning architectures (e.g., LSTM or CNNs), might have leveraged the embeddings more effectively due to their ability to capture complex patterns and relationships in high-dimensional data.
>
> We thank you for this observation. We will experiment with these approaches in our future work.
>
> (3) Table 2 should present in bold the best result for each metric in each predictive horizon.
>
> We thank you for the observation. We have updated the manuscript accordingly.
>
> (4) Typo: sometimes it is written one-hot encoded, other times one hot encoded. It must standardize. On line 316 it just says "hot encoded"
>
> Thank you for the observation. We have updated the manuscript, to ensure consistency.

---

### Official Review · Reviewer_3Ake · 2024-10-01
**Lack of novelty and mathematical justifications**

**Confidence:** 4

**Summary:**

The paper compares different approaches in predicting issues in high-performance computing systems. For this, the authors utilize the M100 dataset consisting of telemetry data with known anomalies. In the experiments, the authors compare 4 different approaches to predict future anomalies by using a discretized version of the time series and incorporating a natural visibility graph.

**Strengths:**

+ Interesting idea of using natural visibility graphs, although their motivation is a bit lacking
+ Detailed and insightful explanation of the dataset
+ Clear overview of the results

**Weaknesses:**

- Transformation of the data lacks justification (e.g., the discretization)
- Although interesting, it's unclear why natural visibility graphs are used instead of simple lagged versions of columns to incorporate time structures
- Seeing that the model is trained for predicting 1, 2, or 3 steps ahead, it is unclear how this is better than the works discussed in the related work section which seem to be applicable in a similar way
- Lack of comparison with existing approaches in the experiments
- No real novelty aside from the particular approach of data processing and chosen modeling approach

**Final Rebuttal Confidence:**

4

**Final Rebuttal Justification:**

I want to thank the authors for their careful answers. While a few of my concerns are addressed, there is still a significant lack of theoretical justification, although I appreciate the empirical work and insights. My biggest concern about the justification of using visibility graphs remains, and I believe a more direct graphical representation (e.g., a (dynamic) Bayesian network or a graphical causal model) would be a much clearer formulation and a more direct modeling of domain knowledge. I strongly recommend looking into related works about graphical causal models as this seems to be a great application for it. While I am willing to increase my score, I cannot give a recommendation for acceptance due to the lack of theoretical work.

**Justification:**

Some remarks and suggestions:

The paper has some interesting ideas but seems rather arbitrary in many of the data processing choices. For instance:

- Why was a forward-filling imputation strategy chosen?
- Why does the data need to be discrete (and why the particular choice of 5 values)?
- Although an interesting approach, why use natural visibility graphs?

These seem to be design decisions that worked the best in practice but lack mathematical justifications. For instance, do they work particularly well with the chosen ML models? Generally, the work lacks any kind of theoretical statements about made assumptions.

Regarding the visibility graphs, wouldn't it be easier to simply take a lagged representation of past timestamps to incorporate temporal dependencies? For instance, to use timestamp t-1, you can simply make a copy of the column and move it by 1 timestamp. This can then be used in your model. For other ideas on how to represent a time series as a (causal) graph, you might want to check out Section 10 in the book Elements of Causal Inference (https://mitpressbookstore.mit.edu/book/9780262037310).

When comparing the related work, you point out that some are not able to give a concrete timeframe to predict future failures. Based on your setup by defining n+1,..., n+3 as your targets, couldn't this also be done in the related works? In this regard, there is also a lack of comparison, e.g., you can try to bring the problem into a similar setting and compare your methodology accordingly as this is the main contribution.

For the training splitting, if you assume that you have a specific order of a Markov process (e.g., the current timestamp t is independent of the t-n timestamps conditioned on timestamps t-1, ..., t-n-1), you actually don't need to ensure that the training data is before the test data as these would be independent segments. However, your strategy makes sense if you assume long-term dependencies (here, n is large).

In the results, Experiment B performed poorly, but there is a lack of explanation. Adding some thoughts or potential reasons could strengthen the insights.

Generally, it's unclear why you don't treat these as time series in the first place and use time series prediction models for this problem (see e.g., https://www.sktime.net/en/stable/)? Some discussion on this would be very insightful.

---

> ### Author Rebuttal · Authors · 2024-10-25
>
> We thank the reviewer for the comments. We provide a point-by-point answer below.
>
>  (1)- Why was a forward-filling imputation strategy chosen?
>
> The forward-fill strategy was chosen under the assumption that if a sensor did not emit a value or if a reading was lost, the most likely value to exist if conditions did not change in between, it would be the latest sensor reading. We have updated the manuscript making this assumption explicit.
>
>  (2)- Why does the data need to be discrete (and why the particular choice of 5 values)?
>
> We observed that the sensor values mostly stayed close to a certain value for prolongued periods of time, signalling a particular state. While small oscilations could be simplified with a change-level detection and replacing the whole range value by its mean, we could still have values that were close to each other at no contiguous segments. Therefore, grouping them into bins would preserve most of the information while discretizing the whole signal into meaningful states. The choice of 5 bins was empirical, considering the amount of states that resulted from such a binning. We have updated the manuscript with the corresponding explanation.
>
>  (3) Although an interesting approach, why use natural visibility graphs?
>
> We have choosen natural visibility graphs as one of the possible heuristics to turn time series into graphs. In particular, we are interested on how time series information could be attached to domain knowledge (e.g., sensors and the kind of measurements they perform) and graphs seemed a natural choice to do so. Therefore, we modelled the sensors and their types as a graph, attaching the corresponding natural visibility graphs that provided insights into measurements from that particular sensor.
>
>  (4) These seem to be design decisions that worked the best in practice but lack mathematical justifications. For instance, do they work particularly well with the chosen ML models? Generally, the work lacks any kind of theoretical statements about made assumptions.
>
> We thank you for this observation. We have updated the manuscript to emphasize the rationale behind the data preprocessing and design decisions regarding evaluating graph representations for time series.
>
>  (5)- Regarding the visibility graphs, wouldn't it be easier to simply take a lagged representation of past timestamps to incorporate temporal dependencies? For instance, to use timestamp t-1, you can simply make a copy of the column and move it by 1 timestamp. This can then be used in your model. For other ideas on how to represent a time series as a (causal) graph, you might want to check out Section 10 in the book Elements of Causal Inference (https://mitpressbookstore.mit.edu/book/9780262037310).
>
> We thank you for this observation. While including lagged representations of past timestamps would capture sensor data, we would still lack information regarding domain knowledge such as which sensor originated the sensor values and the kind of variable the sensor is measuring (e.g., temperature, power, etc.). Such information can be encoded as a graph, providing additional insight to the model regarding the sensors within the racks and data center. Using approaches such as natural visibility graphs to convert time series into graphs allows us to convey the time series information to the graph and capture it all together with an embedding. We consider a graph representation to be key for capturing information such as sensor IDs, their type, and placement - information that would otherwise be difficult to include in the model.
>
> We thank you for the reference regarding causality and time series. While we were not able to explore it for this research, we will for sure explore it in the future.
>
> (6) When comparing the related work, you point out that some are not able to give a concrete timeframe to predict future failures. Based on your setup by defining n+1,..., n+3 as your targets, couldn't this also be done in the related works? In this regard, there is also a lack of comparison, e.g., you can try to bring the problem into a similar setting and compare your methodology accordingly as this is the main contribution.
> We thank you for the comment provided above. Yes, we can provide an estimate of the time ahead we are able to forecast with each of the time horizons. In particular, we have seen that the states changed every 165 minutes on average and that we could be predicting at most up to 495 minutes ahead. The state of the art predictions are made by the GRAFFE model with a four-hour look-ahead window.
>
> (7) For the training splitting, if you assume that you have a specific order of a Markov process (e.g., the current timestamp t is independent of the t-n timestamps conditioned on timestamps t-1, ..., t-n-1), you actually don't need to ensure that the training data is before the test data as these would be independent segments. However, your strategy makes sense if you assume long-term dependencies (here, n is large).
>
> We thank you for this observation. You are right. We are keeping the setup in this way, so that we can compare our results with additional models we aim to develop in the future, which will be considering sequences of states e.g., through temporal graphs or similar approaches.
>
> (8) In the results, Experiment B performed poorly, but there is a lack of explanation. Adding some thoughts or potential reasons could strengthen the insights.
>
> We thank you for this comment. We have updated the results section with additional explanations. We consider several hypotheses but have no certain answer to them yet, and further research is required to reach a better understanding: (i) the natural visibility graphs capture only the topological aspects of the time series, and any information related to their particular values is lost. We could amend this void by adding complementary graph representations such as quantile graphs or ordinal partition graphs; (ii) the quality of the Graph2Vec embeddings depends much on how the graph was sampled - we need to explore how different hyperparameters affect the outcomes (e.g., number of paths and their length).
>
> (9) Generally, it's unclear why you don't treat these as time series in the first place and use time series prediction models for this problem (see e.g., https://www.sktime.net/en/stable/)? Some discussion on this would be very insightful.
>
> In particular, we are interested on how time series information could be attached to domain knowledge (e.g., sensors and the kind of measurements they perform) and graphs seemed a natural choice to do so. Therefore, we modelled the sensors and their types as a graph, attaching the corresponding natural visibility graphs that provided insights into measurements from that particular sensor. We have updated the manuscript with this clarification.
>
>
> We would be grateful for the reviewer to reconsider their score based on the updates described above and the new version of the manuscript we have uploaded.

---

### Official Review · Reviewer_jTLt · 2024-10-07
**The study focuses on predicting compute node failures in HPC clusters with learning form graph approach.**

**Confidence:** 3

**Summary:**

The study compares four machine learning approaches for predicting compute node downtime, three of which are based on graph embeddings, and compares the results.

**Strengths:**

Feature extraction methods based on proposed graph embeddings

**Weaknesses:**

Poor writing of the study
The abstract does not reflect the study and does not contain the information required in an academic abstract.
The introduction is poorly written and does not include a paragraph about the contribution of the study.
Failure to explain all the steps taken in the study with good examples.
Poor placement of Figures and Tables in the work

**Final Rebuttal Confidence:**

3

**Final Rebuttal Justification:**

The authors still could not provide the desired academic article summary, especially in the abstract. For example, the contribution of the work is not quantified. Success is expressed in vague words. Again, unfortunately, there is no clear statement in the introduction for a clear contribution to the literature. Despite all these, I can say that it is an acceptable study.

**Justification:**

The study was evaluated by considering all the points that should be in an academic study, starting from the summary to the conclusion. Although it is stated that it contains an innovation, it is not clearly emphasized in this study and the work done could not be expressed in a good way.

---

> ### Author Rebuttal · Authors · 2024-10-25
>
> We thank the reviewer for the comments provided above. We have reworked the whole paper, improving the abstract, introduction, and part of the related work. We have also updated the sections related to the methodology, experiments, and results, making explicit the research questions we attempt to answer with our research work, providing insights into the rationale behind the preprocessing steps and experiments, and analyzing the results obtained in more depth. We have also updated the Figures, not only providing a better placement but replacing some of them with more informative ones to ensure the reader gets a better understanding of the work performed. We would be grateful for the reviewer to reconsider their score based on the updates described above and the new version of the manuscript we have uploaded.

---

### Meta-Review · Area_Chair_oAFF · 2024-10-31

**Recommendation:** Accept (Oral)
**Confidence:** 4

**Metareview:**

The paper presents a comparative analysis of various approaches for predicting issues in high-performance computing (HPC) systems. Strengths of the work include a detailed and insightful explanation of the dataset and a clear overview of the results. A key weakness, however, is the limited discussion on the experimental results, although the proposed methodology is sufficiently innovative. Additionally, the motivation behind the study is not sufficiently clear, and the introduction lacks background information, making it challenging for readers outside the domain to grasp the context and significance of the research.

Following the rebuttal, the authors have significantly improved the paper by adding background information and additional experimental insights. They have also refined their discussion of the results, making it more comprehensive and appropriately detailed.

Considering all comments from the reviewers, as well as the authors' constructive responses, I would recommend acceptance of the paper.

**Suggested Changes To The Recommendation:**

1: I agree that the recommendation could be moved down

---

### Decision · Program_Chairs · 2024-11-06

**Decision:**

Accept (Oral)

**Comment:**

We recommend an oral and a poster presentation given the AC and reviewers recommendations.